# PRETRAINED MODELS ARE GOOD ACTIVE LEARNERS

## ABSTRACT

An important barrier to the safe deployment of machine learning systems is the risk of *task ambiguity*, where multiple behaviors are consistent with the provided examples. We investigate whether pretrained models are better active learners, capable of asking for example labels that *disambiguate* between the possible tasks a user may be trying to specify. Across a range of image and text datasets with spurious correlations, latent minority groups, or domain shifts, finetuning pretrained models with data acquired through simple uncertainty sampling achieves the same accuracy with **up to 6× fewer labels** compared to random sampling. Moreover, the examples chosen by these models are preferentially minority classes or informative examples where the spurious feature and class label are decorrelated. Notably, gains from active learning are not seen in unpretrained models, which do not select such examples, suggesting that the ability to actively learn is an emergent property of the pretraining process.

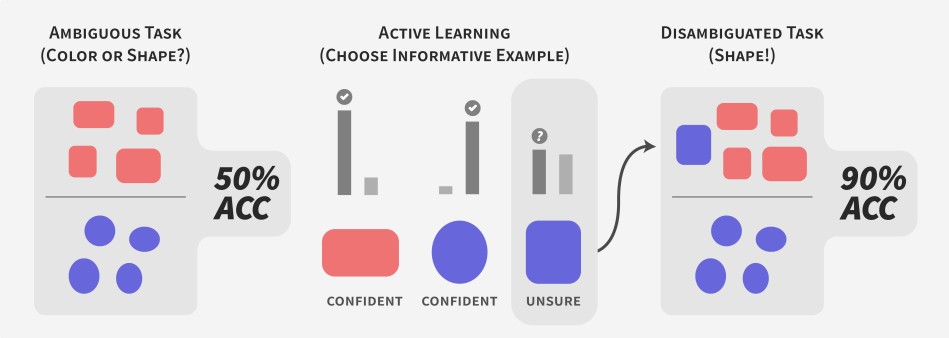

Figure 1: **Active learning can resolve task ambiguity, which is especially salient in few-shot settings.** Here, the provided training data leaves the model unsure of the task: is it to predict the shape or the color of the object? Pretraining enables models to disentangle and weigh various competing features, making them good active learners that can choose disambiguating examples (e.g. the blue square), resolving this task ambiguity.

## 1 INTRODUCTION

Modern pretrained models can be adapted to new tasks with remarkably little data, enabling downstream applications for tasks with only tens or hundreds of examples (Brown et al., 2020; Radford et al., 2021). However, an important but neglected challenge that is especially salient in few-shot settings is *task ambiguity*, when the desired behavior is not uniquely specified by the provided examples. Task ambiguity can manifest in different ways: For example, the class-relevant features of an input (e.g., the shape of an object in Figure 1) may be spuriously correlated with other features predictive of the training labels (e.g., the color of the object), making the desired task unclear. In addition, task ambiguity may arise from an underdiverse training set, causing models to be unsure of the desired behavior for minority groups or during distribution shifts that occur at test-time.

We consider whether the task ambiguity problem can be addressed through *active learning*, where models select informative examples for users to label. In principle, active learning allows models themselves to assist in resolving task ambiguity by identifying examples whose labels would be

informative; for example, in fig. 1, asking for the label of the blue square helps determine that the desired task is to predict the object shape not the object color. Since it may be challenging in general for users to intuit possible sources of task ambiguity, much less how to address them with example selection, an automated active learning approach may be desirable.

In traditional settings with smaller, unpretrained models, several challenges often prevent active learning from seeing success in practice, including label noise, unlearnable examples, and a lack of generalizability across models and tasks (Lowell et al., 2019; Karamcheti et al., 2021). However, using pretrained models in few-shot or low-data settings has several key differences from traditional ML pipelines which may make them particularly well-suited for active learning. First, the effect of an individual data point (and the difference between an informative vs. an uninformative one) is magnified, compared to the thousands or millions of examples typically acquired through standard active learning pipelines. Second, pretrained models excel at learning high-level representations of inputs, which better surface relevant features and may encourage active learning to select examples that disambiguate between these high-level features (e.g. shape and color).

We consider the use of active learning on a range of spuriously correlated, imbalanced, or domain shifted datasets where task ambiguity is salient. We compare an active learning approach with a random-sampling baseline, and compare the difference in performance with and without the use of pretrained models. Our contributions are:

1. Identifying and motivating task ambiguity as a unified concern across a range of real-world datasets, and an especially salient problem for low-data settings.

2. Proposing active learning as a potential solution, along with a simple yet practical recipe that does not require adjustment for new datasets or tuning on validation data.

3. Showing that active learning can enable large gains (up to $6\times$ reduction in data points, +12% absolute gain for the same labeling budget) and presenting scaling trends demonstrating that this is an emergent property of pretraining.

## 2 METHOD

We study the pool-based active learning setup common in the literature (Settles, 2009), where we have a (possibly pretrained) model $\mathcal{M}$, a small *seed set* of training data $\mathcal{S} = \{(x_i, y_i)\}$, and a larger *pool* of unlabeled data $\mathcal{P} = \{x_i\}$. The active learning procedure proceeds as follows: first, finetune $\mathcal{M}$ on $\mathcal{S}$ until convergence; then, select points $x_i$ from $\mathcal{P}$ that are deemed *most informative* according to an acquisition function $a(x; \mathcal{M})$, obtaining the corresponding labels $y_i$, until some budget $k$ of data points is exhausted. This newly labeled batch $\mathcal{B} = \{(x_i, y_i)\}$ is then added to the existing data $\mathcal{S}$, and the model is retrained on $\mathcal{S}$ for the next acquisition step. This process is repeated for a fixed number of acquisition steps.

For our acquisition function, we adopt the classic *uncertainty sampling* approach to active learning, in particular the *least confidence* heuristic, where we acquire points for which our model is *least confident* in its predicted label (Settles, 2009). Specifically, treating the outputs of the model $\mathcal{M}$ as a probability distribution[1] over possible labels $p(y \mid x; \mathcal{M})$, we define the acquisition function to be

$$a(x; \mathcal{M}) = -\max_i p(y_i \mid x; \mathcal{M}) \tag{1}$$

Although we could use more complex measures such as the *entropy* over labels, the *least confidence* heuristic has shown to be simple and effective in a variety of settings (Settles, 2009; Hendrycks & Gimpel, 2017; Mussmann et al., 2020), and we similarly find good results here (we refer to least confidence sampling as "uncertainty sampling" except in Section 4.3 where we explore other uncertainty-based acquisition functions).

On top of this standard active learning pipeline, we propose the following change to improve the practical applicability of pretrained models in data-scarce settings:

---

[1]While in general there is no guarantee that this probability distribution will be well-calibrated, recent work has found that pretraining improves model calibration across a variety of settings, including on out-of-domain data (Desai & Durrett, 2020; Hendrycks et al., 2019).

**Removing the need for a separate validation set**   The active learning cycle begins by finetuning the pretrained model on the seed set. If the size of the seed set is large enough, the seed set may be partitioned into a training set and a validation set, and early stopping may be performed on the validation set. However, in few-shot settings, labeling costs may be high, and the seed set may be too small to meaningfully partition. This has led to a variety of ad-hoc solutions, e.g. finetuning for an arbitrary constant number of gradient steps depending on the task (Kolesnikov et al., 2020).

Instead of an arbitrary fixed number of finetuning steps, we propose an alternative method to terminate finetuning in the absence of a validation set. Specifically, we found that a simple but effective heuristic was to stop finetuning when the training loss decreases to 0.1% of the original training loss at the start of finetuning. In our experiments, this heuristic performed as well as early stopping on an actual validation set (see Appendix D for more details). By using a standardized recipe across tasks and removing the need for a separate validation set, our active learning pipeline is more robust to the real-world difficulties of deploying active learning in data-scarce settings where use of a validation set is impractical (Lowell et al., 2019; Perez et al., 2021), although further work is needed to capture the full extent of this recipe's generalizability.

## 3   DATASETS

We consider a variety of datasets where task ambiguity is especially likely when only a few examples are provided, especially due to spurious correlations, latent minority classes, and domain shifts. These datasets provide an empirical testbed for the ability of pretrained models to choose disambiguating examples using active learning.

### 3.1   DISTINGUISHING CAUSAL FROM SPURIOUS FEATURES

Spurious correlations arise when multiple features are predictive of the label in a training dataset, yet it is ambiguous which ones are causally linked to the task label (Geirhos et al., 2020). We consider two such datasets, and see whether active learning can choose the *disambiguating examples* where the spurious features are not copresent with the causal features:

**Waterbirds**   The Waterbirds dataset (Sagawa et al., 2019) consists of photographs of landbirds or waterbirds digitally edited onto land or water backgrounds. The task is to classify whether the bird is a landbird or a waterbird. In the train set, 77% of the pictures feature landbirds and 23% waterbirds. 95% of both landbirds and waterbirds appear on land and water backgrounds, respectively. In the validation and test sets, this percentage is decreased to 50%, instead. Thus, the image background is a spurious feature the model may come to rely on when making the prediction.

**Treeperson**   As the Waterbirds dataset was synthetically generated, we also consider a dataset where we perform classification over real, unedited images with spuriously correlated objects. We use the object annotations in Visual Genome (Krishna et al., 2016) to create a new dataset of 8,638 images called Treeperson, for which the task is to predict whether a person is in a given image. While 50% of the images contain a person in this dataset, each image also contains either a tree or a building, and the presence of these objects is spuriously correlated with the presence of people. At train time, 90% of training images with people contain a building, while 90% of training images without people contain a tree. Thus, a model may be incentivized to form representations that classify according to the presence of trees and buildings, rather than the presence of the actual causal variable of interest (people). These values are changed to 50% at test time, removing this correlation to evaluate how well the model learned the actual feature of interest. For more details on this dataset, see Appendix C.

### 3.2   MEASURING ROBUSTNESS TO DISTRIBUTION SHIFT

Distribution shifts occur when algorithms are evaluated on different data distributions than the ones they were trained on. Examples include changing the location or time of day that photos were taken, or changing the topic or author of a particular textual source. These shifts can reduce performance, and we consider whether active learning can help choose diverse, informative examples that clarify how the model should behave over a range of natural distribution shifts.

**iWildCam2020-WILDS**  This dataset considers the task of species classification from a database of photos taken from wildlife camera traps (Beery et al., 2020; Koh et al., 2021). The dataset is unbalanced, with most images containing no animal, and the distribution of camera locations and species changes between the in domain (ID) and out-of-domain (OOD) subsets.

**Amazon-WILDS**  This dataset considers the task of predicting the number of stars associated with the text of a given Amazon review (Ni et al., 2019; Koh et al., 2021). The reviewers are different in the training set versus the test set, and the task is to perform as well as possible on this set of new reviewers. In addition to number of stars, we also consider model performance stratified by different product types, which highlights minority subgroups whose categorization is not visible to the model.

## 4  EXPERIMENTS

### 4.1  MODELS AND TRAINING

**Vision**  For computer vision datasets, we finetune BiT (Kolesnikov et al., 2020), a recently-proposed family of vision models which have achieved state-of-the-art performance on several vision tasks. We primarily consider the BiT-M-R50x1 model, pretrained on ImageNet-21k (Deng et al., 2009). To explore the effectiveness of larger architectures and pretraining sources, in Section 5.2 we also consider performance achieved by the same-size BiT-S-R50x1, trained on ImageNet-1k, and the deeper BiT-M-R101x1 model, also trained on ImageNet-21k. These models have been shown to have emergent few-shot learning abilities, where strong classifiers for new tasks can be obtained by simply finetuning on tens or hundreds of examples with typical gradient descent techniques (rather than meta-learning techniques, for example).

**Text**  For the text dataset (Amazon), we use RoBERTa-Large (Liu et al., 2019), another pretrained model with similar properties as BiT, and a representative of the BERT (Devlin et al., 2019) family of models which together have obtained state-of-the-art scores on modern NLP benchmarks (Wang et al., 2018).

Other details, including hyperparameters and seed set/acquisition sizes are deferred to Appendix B.

**Random acquisition baseline**  As a running baseline, we compare to the same model finetuned with a *random acquisition* function (equivalent to not doing active learning). That is, $a(x; \mathcal{M}) = \text{rand}(0, 1)$, so we simply sample a random batch of data from the pool at each acquisition step.

**Comparison with unpretrained models**  To examine whether effective active learning is an emergent property of the pretraining process, we also compare to the performance observed when applying active learning to a *randomly initialized*, instead of pretrained, BiT-M-R50x1.

### 4.2  ACCURACY PER ACQUISITION

For a general measure of success, we plot the accuracy of active learning versus random sampling on the validation datasets as a function of the number of samples acquired during training.

**Waterbirds**  Waterbirds is evaluated on a balanced dataset where the foreground and background are not correlated. In this setting, uncertainty sampling achieves a **+12% improvement** in average validation accuracy over random sampling (Figure 2a). This comes primarily from a **+31% average increase** across the landbird-on-water and waterbird-on-land images (i.e. those without the spurious correlation; Figure 2b). Uncertainty sampling required **5.8x fewer labels** than random sampling to achieve random sampling's final accuracy.

**Amazon**  In the Amazon dataset, we also see gains from active learning in the OOD setting, including **+1% on average** across reviewers, and **+2.5% on the worst 10th percentile** (Figure 2g). This result suggests that our active learning recipe may be of use outside of BiT or computer vision settings more broadly. The confidence not overlapping by the end of training is evidence that

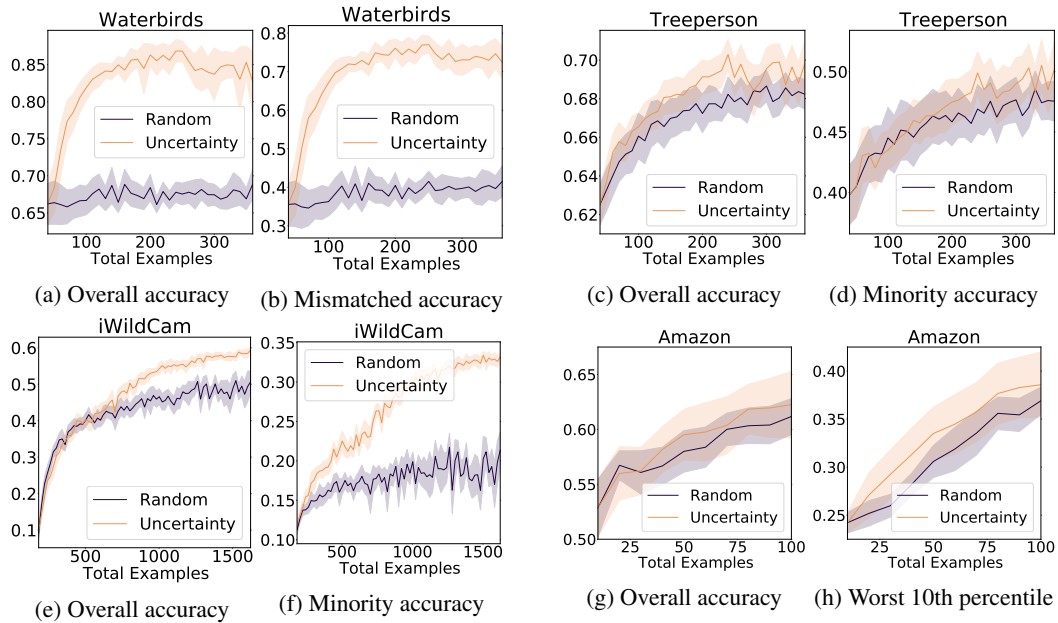

Figure 2: **Uncertainty sampling outperforms random sampling on all datasets, especially on minority classes.** Shaded regions represent 95% CIs (Gaussian approx.).

while the difference between uncertainty and random sampling is not large, it is statistically significant. Uncertainty sampling required **1.3x fewer labels** than random sampling to achieve random sampling's final accuracy.

**iWildCam** With the iWildCam dataset, uncertainty sampling achieved a **+8% improvement** upon random sampling. Uncertainty sampling also required **1.9x fewer labels** than random sampling to achieve random sampling's final accuracy (Figure 2g).

**Treeperson** In the Treeperson dataset, uncertainty sampling is **+2% improved** over random sampling by the end of training (Figure 2c). Uncertainty sampling required **2.5x fewer labels** than random sampling to achieve random sampling's final accuracy.

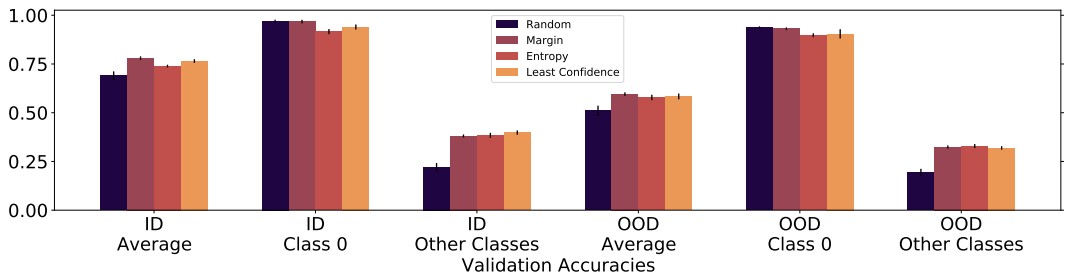

Figure 3: **All types of uncertainty sampling outperform random sampling on iWildCam.**

## 4.3 ADDITIONAL ACTIVE LEARNING METHODS

We consider two additional active learning methods in addition to least confidence sampling: 1) entropy sampling, which chooses the example that maximizes the entropy of the model's predictive distribution, and 2) margin sampling, which chooses the example with the smallest difference between the first and second most probable classes (Scheffer et al., 2001; Settles, 2009). We run

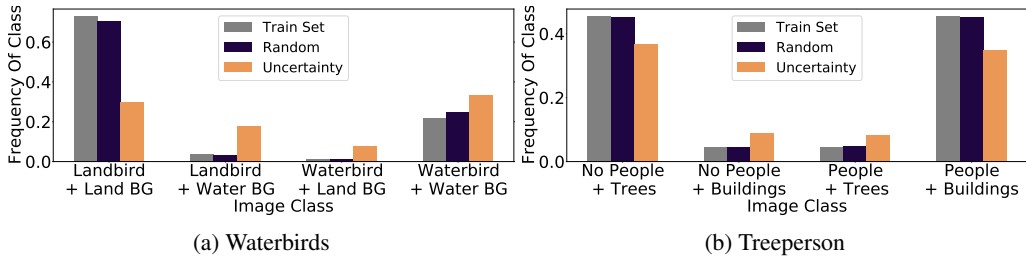

(a) Waterbirds          (b) Treeperson

Figure 4: **Uncertainty sampling identifies and upsamples disambiguating examples.** For both Waterbirds and Treeperson, uncertainty sampling selectively acquires examples where the spurious and core features disagree. Y-axis: percentage oversampling of uncertainty over random sampling.

experiments with all methods on the 182-class iWildCam dataset[2]. All methods significantly outperform random sampling (Figure 3).

### 4.4 ADDITIONAL PRETRAINED VISION MODEL

We broaden our coverage of computer vision models to include vision transformers (Dosovitskiy et al., 2021), the other major architecture family currently in use. We train ViT-16/B,[3] on Treeperson, observing smaller gains for AL than the BiT model (Figure 8). This perhaps reflects the fact that the vision transformer was pretrained on ImageNet-21k for far fewer epochs then BiT (9 vs 70), and is corroborated by much lower oversampling of minority classes than BiT (Figure 9).

## 5 ANALYSIS

### 5.1 ACTIVE LEARNING SELECTS EXAMPLES THAT RESOLVE TASK AMBIGUITY

Overall, we attribute improved performance to pretrained models' ability to identify and preferentially sample disambiguating examples and latent minority examples that resolve task ambiguity.

**Waterbirds** Figure 4a depicts the rate at which uncertainty sampling is acquiring examples of each subgroup compared to the expected rate at which random sampling would acquire examples from those same subgroup. Examples where the bird and background are mismatched are heavily oversampled. We emphasize that these minority examples are not simply members of the minority *class* (waterbirds). If active learning were simply upsampling minority classes, then all landbird images would be downsampled and all waterbird images would be upsampled. However, this is not the case—instead, the model identifies and preferentially upsamples *informative* examples where the spurious feature (background) and the causal feature (bird type) disagree.

**Treeperson** For Treeperson we see the same pattern as in Waterbirds: the model identifies and upsamples examples where only one of the spurious or causal features is present (Figure 4b).

**Amazon** We also see similar behavior in the Amazon dataset, indicating our method's applicability to multiple modalities and pretrained models. Not only does the model upsample lower star ratings, which are less common, it is also able to upsample rarer product categories—an unseen attribute.

### 5.2 PRETRAINING IS CRUCIAL FOR ACTIVE LEARNING TO WORK

We also consider the role of the pretraining dataset and model size. We conduct active learning experiments with 3 pretrained models: BiT-S-R50x1, BiT-M-R50x1, BiT-M-R101x1, and their corresponding non-pretrained versions. For all pretrained models (except BiT-M R-101 for Treeperson)

---

[2]Note that least confidence, entropy, and margin sampling are identical in the case of binary classification.
[3]https://huggingface.co/google/vit-base-patch16-224

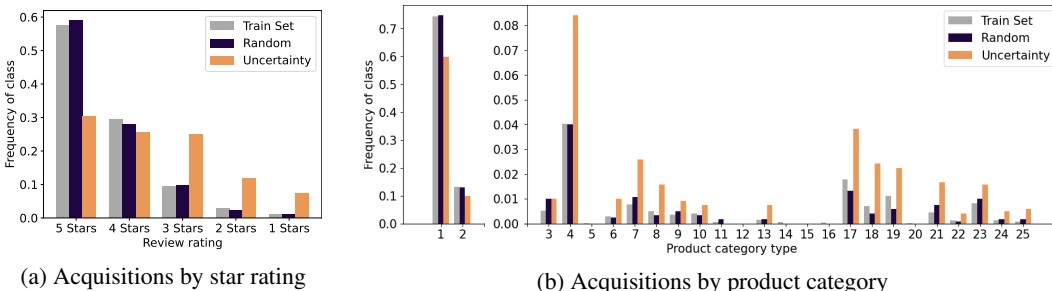

(a) Acquisitions by star rating

(b) Acquisitions by product category

Figure 5: **Uncertainty sampling upsamples both visible and latent minority subgroups.** Fraction of Amazon examples acquired by random and uncertainty sampling, stratified by star rating and product category. Uncertainty sampling preferentially acquires examples with lower star ratings and rarer product categories, despite the latter attribute not being visible to the model.

uncertainty acquisition outperformed random acquisition (Figure 7a, Figure 7b, Figure 7c). Importantly, we did not see any gains from AL on the unpretrained models, even when exploring a range of different hyperparameter configurations.

**Scaling trend**  Interestingly, the BiT-S-R50x1 model, which was pretrained on a smaller dataset than the BiT-M models (ImageNet-1k vs ImageNet-21k) achieves less of a boost from active learning on iWildCam vs the other two models. This suggests that the gains from active learning may continue to grow as pretained models are trained for longer on more data. However, we did not see a difference between BiT-M-50x1 and BiT-M-101x1, which were trained on the same dataset but have different numbers of parameters. We encourage future work that more thoroughly examines these behaviors and scaling laws across a range of models and datasets.

**Impact of pretraining on acquisition patters**  Pretrained models also acquire disambiguating subgroups much more efficiently than their unpretrained counterparts. See Appendix G for additional figures and results illustrating this point.

## 5.3  PRETRAINING PROVIDES A BETTER FEATURE SPACE FOR ACTIVE LEARNING

Why does pretraining improve AL for resolving task ambiguity? We investigate the hypothesis that pretrained representations enable AL models to select objects based on higher-level features (e.g. object foreground and background) through various analyses on Waterbirds.

**Linear classifier analysis**  We investigate the presence of these high-level features by training linear classifiers on the second to last layer of BiT models. The classifiers are trained to predict each image's bird type and background type (4 classes, each comprising 25% of the data). As shown in Figure 6, these classes are more linearly separable in pretrained models both before any finetuning, as well as after finetuning on a seed set and then acquired examples from the original 2-class Waterbirds dataset. This demonstrates that pretrained models can better identify both the causal and spurious features in the data, potentially explaining why they can choose them preferentially.

| | No Finetuning | Finetune On Seed Set (40) | Finetune On Seed Set (40) + 20 |
|---|---|---|---|
| Average | 0.402 | 0.42 | 0.424 |
| Landbird /LandBG | 0.389 | 0.416 | 0.42 |
| Waterbird /LandBG | 0.63 | 0.653 | 0.733 |
| Landbird /WaterBG | 0.426 | 0.467 | 0.447 |
| Waterbird /WaterBG | 0.435 | 0.418 | 0.419 |

| | No Finetuning | Finetune On Seed Set (40) | Finetune On Seed Set (40) + 20 |
|---|---|---|---|
| Average | 0.311 | 0.32 | 0.34 |
| Landbird /LandBG | 0.306 | 0.321 | 0.369 |
| Waterbird /LandBG | 0.194 | 0.316 | 0.325 |
| Landbird /WaterBG | 0.286 | 0.248 | 0.262 |
| Waterbird /WaterBG | 0.337 | 0.33 | 0.255 |

(a) Group accuracies for linear classifier on Waterbirds image embeddings attained from a pretrained BiT model after various degrees of finetuning

(b) Group accuracies for linear classifier on Waterbirds image embeddings attained from an unpretrained BiT model after various degrees of finetuning

Figure 6: **Both causal and spurious features are more linearly separable in pretrained models.**

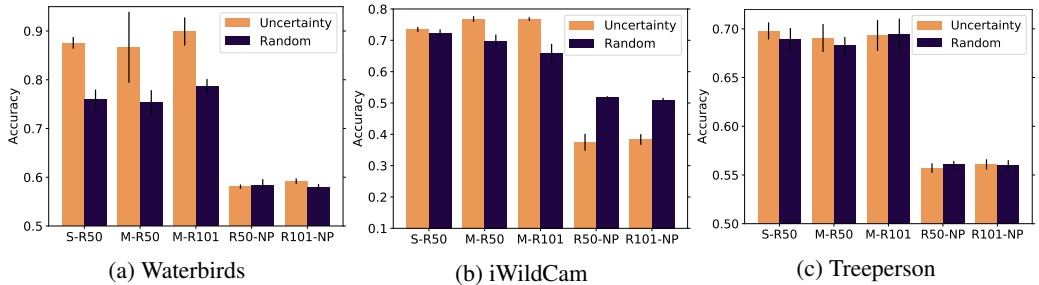

Figure 7: **Uncertainty sampling only provides gains when using pretrained models.** S-R50, M-R50, and M-R101 correspond to the BiT-S-R50x1, BiT-M-R50x1, and BiT-M-R101x1 pretrained models, respectively, while R50-NP and R101NP correspond to ResNet models which are not pretrained. Shaded regions represent 95% CIs (Gaussian approx.).

**t-SNE visualization** We visualize the second-to-last layer embeddings of BiT (without any fine-tuning) using t-SNE (van der Maaten & Hinton, 2008), shown in Figure 13. We find that examples selected by uncertainty sampling fall closer to the decision boundary for the pretrained model than for the unpretrained model. We also see that the pretrained model exhibits far more separation by class, suggesting it is better able to identify useful features in the input. These conclusions help further characterize how pretraining provides a feature space more amenable to active learning.

### 5.4 PINPOINTING THE EFFECT OF TASK AMBIGUITY WITH A DOSE-RESPONSE EXPERIMENT

To examine the impact of task ambiguity from the perspective of the unlabeled dataset, we construct variants of the waterbirds dataset where the percentage of mismatched examples range from 95% to 50%. [4] We then proceed with AL and report results in Figure 10.[5] We find significantly larger gains on versions of the dataset with smaller fraction of mismatched examples (gains average 4% for 50–70% shift, and 10% for 75–95% shift) keeping. This provides stronger evidence that active learning is actually helping the model identify the true task.

### 5.5 FAILURE CASES

To assess how similar the datasets needed to be to the training distribution of BiT, we also perform preliminary experiments on Camelyon17-WILDS (Bándi et al., 2019; Koh et al., 2021), which considers tumor identification from tissue patches, and FMoW-WILDS (Christie et al., 2018; Koh et al., 2021), which considers land-use classification from satellite images. Active learning performs comparably or worse than random sampling on these datasets, even when using a pretrained BiT model, suggesting that generalization to domains far from the training distribution (ImageNet-21k) may be challenging. However, Camelyon17-WILDS is also known to exhibit high variance across seeds,[6] which may also be a contributing factor.

## 6 RELATED WORK

**Task ambiguity and specification** Several works address ambiguity or poor specification in machine learning problems. Taylor et al. (2020) describe the problem of "inductive ambiguity identification," and describe active learning as a promising potential solution that has failed to see practical success. D'Amour et al. (2020) describe the problem of *underspecification*, where high variance, instability, and poor model performance result from training overparameterized models on small amounts of data. Geirhos et al. (2020) describes how task ambiguity can arise when both desirable and undesirable features are predictive of the training labels, a problem which several works seek to better characterize and address (Nagarajan et al., 2021; Sagawa et al., 2019; 2020). Finally, Finn et al. (2018) address task ambiguity in few-shot settings via a probabilistic meta-learning algorithm,

---

[4]We construct these datasets using the code at `https://github.com/kohpangwei/group_DRO`.

[5]Results are preliminary and will be rerun with more seeds.

[6]`https://wilds.stanford.edu/get_started/`

and perform an active learning experiment in a 1D regression setting. We build on these works by demonstrating that simple uncertainty sampling with pretrained models can be an effective approach to the task ambiguity problem across a wide variety of high-dimensional classification settings.

**Uncertainty and distribution shift**   In the face of these challenges, several works have tried to quantify how much pretrained models know about problems or their own uncertainty about them. Rajpurkar et al. (2018) propose a question answering dataset with unanswerable questions, where a model must abstain rather than proceeding with an answer. Pretraining can also improve the calibration of model uncertainty (Hendrycks et al., 2019) and pretrained features can be used for out-of-distribution detection (Reiss et al., 2021; Wu & Goodman, 2020)—observations that align with our findings that uncertainty sampling can identify minority subgroups in datasets. Furthermore, our observation that upsampling latent minority groups results in better performance aligns well (Sagawa et al., 2020), which found that simply upweighting minority groups performed less well than increasing the relative fraction of minority group examples in the distribution. Importantly, however, our active learning setup does not require these groups to be known in advance.

**Active learning and example selection**   Active learning (AL) (Lewis & Catlett, 1994; Settles & Craven, 2008; Settles, 2009; Houlsby et al., 2011; Aggarwal et al., 2020) is a well-studied field that investigates how machine learning algorithms might automatically select helpful additional data points to maximize their performance. Such strategies are especially helpful in imbalanced settings (Ertekin et al., 2007; Mussmann et al., 2020) and has been fruitfully applied to deep models (Gal et al., 2017; Beluch et al., 2018), including pretrained models (Yuan et al., 2020; Margatina et al., 2021; Shelmanov et al., 2021). Past work has also considered AL for few-shot learning (Woodward & Finn, 2017). We extend these works by specifically considering AL for resolving task ambiguity, showing that AL can induce something closer to the true task desired by users by selecting examples from *unlabeled* minority subgroups, as well as examples that disambiguate causal from spurious features. In contrast to prior work, we also demonstrate the causal effect of pretraining on AL by running controlled studies that compare to both random sampling and unpretrained models.

**Pretrained models and their emergent properties**   Our work contributes to a broader literature on how pretraining enables new kinds of model capabilities (Bommasani et al., 2021; Tamkin et al., 2021), especially when deployed in few-shot settings. For example, (Brown et al., 2020) identify the phenomenon of in-context learning, where tasks can be specified for models through a language modeling prompt, while Caron et al. (2021) discover that a self-supervised vision model implicitly learns high-quality segmentation maps visible through attention scores. Kaplan et al. (2020); Henighan et al. (2020) conduct scaling laws experiments which chart how capabilities emerge with scale. We identify a new model capability that emerges through pretraining: the capacity to actively learn and resolve task ambiguity in conceptually abstract hypothesis spaces.

## 7   DISCUSSION AND LIMITATIONS

We argue that pretrained models are good active learners, capable of identifying informative examples across a diverse range of settings where task ambiguity makes choosing examples challenging. Despite past work largely considering each separate problems, we find that active learning helps in cases where data is spuriously correlated, undergoes domain shift, or contains unlabeled subpopulations. These behaviors emerge most clearly as a result of large-scale pretraining, suggesting that active learning may be an underappreciated tool for increasing the reliability of systems in real-world settings. Importantly, pretraining appears necessary for these behaviors to manifest—and in some cases, models pretrained for longer on larger datasets appear to actively learn better.

Our method does suffer from a number of limitations. First, it requires a human in the loop for data acquisition, which significantly increases the time required to train a model compared to random sampling—a cost which must be weighed against potential benefits. Second, our method requires the labeling method to be relatively free of noise—this may be acceptable if annotators are domain experts or are well-trained, but may also increase the cost per acquired example.

Finally, we note the opportunity for much exciting future work, including deeper investigation of task ambiguity in real-world settings and better understanding how pretraining shapes active learning as models continue to scale.

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

## A  CODE RELEASE

Our code and training scripts will be released at `https://redacted`.

## B  ADDITIONAL EXPERIMENTAL DETAILS

Here we provide some additional experimental details.

All BiT runs use the same default settings specified in the BiT paper (Kolesnikov et al., 2020) and the BiT GitHub repo: `https://github.com/google-research/big_transfer`

Settings that are constant across all BiT runs:

1. **Optimizer** - SGD with momentum 0.9
2. **Learning Rate** - Base learning rate 0.003. Linear warm up to this rate, then staircase decay. Exact schedule depends on dataset size, but for our few shot setting, this means: (1) linear warm up in the first 100 steps to 0.003, then (2) decay 10 fold every 100 steps; (3) after 500 steps, stop and move on to the next acquisition round.
3. **Data Augmentation** - Random cropping and flipping. See our `repo/utils/datasets/load` for details. Also available in BiT repo.
4. **Batch Size** - 32 when training, split into gradient accumulation microbatches of size 8.
5. **Early stopping condition** - When the training loss reaches below 0.001 times the original training loss.

Settings that varied between image datasets:

1. **Size of Initial Seed Set** - 40 for Waterbirds, 40 for Treeperson, 182 for iWildCam, 20 for CIFAR10, 8 for Camelyon17
2. **Size of Training Pool** - Entire training set for Waterbirds, Treeperson, Camelyon17. A random subset of 5000 examples and 12000 examples (re-drawn for each acquisition) from the entire training set of CIFAR10 and iWildCam, respectively.
3. **Number of Acquired Examples** - 320 for Waterbirds, 320 for Treeperson, 1456 for iWild-Cam, 180 for CIFAR10, 128 for Camelyon17

4. **Number of Examples Acquired Each Acquisition** - 5 for Waterbirds, 5 for Treeperson, 20 for iWildCam, 2 for CIFAR10, 1 for Camelyon17.

Settings used for Amazon + RoBERTa-Large runs:

1. **Optimizer** - AdamW with default hyperparameters ($\beta_1 = 0.99, \beta_2 = 0.999$, weight decay = 0.1).

2. **Learning Rate** - LR = 1e-6.

3. **Batch Size** - 2 (due to memory considerations).

   **Early Stopping Condition**   When the training loss reaches below 0.001 times the original training loss or when 4000 gradient steps have been taken (whichever comes first).

4. **Size of Initial Seed Set** - 10.

5. **Size of Training Pool** - A random subset of 2000 examples (re-drawn for each acquisition) from the entire training set.

6. **Number of Acquired Examples** - 90.

7. **Number of Examples Acquired Each Acquisition** - 2.

## C   TREEPERSON DATASET

The Treeperson dataset is composed of images from Visual Genome (Krishna et al., 2016) with different compositions of detected objects.

Training set composition by subclass:

- Person and Building: 3700
- Person and Tree: 370
- No Person and Building: 370
- No Person and Tree: 3700

The validation set contains 498 examples of each subclass.

The following annotated objects were used to form the different subclasses:

- Person: person, people, man, men, woman, women
- Building: building, buildings
- Tree: tree, trees, leaf, leaves, grass

The training set and validation set were drawn randomly from qualifying images in Visual Genome's training set and validation set, respectively.

## D   EARLY STOPPING CONDITION

At the outset of this work, we explored if we could identify a heuristic for stopping training when there was no validation set present. We compared how the BiT model would perform if it stopped the finetuning step based off of when the validation accuracy plateaued versus when the training loss decayed to be 0.001 of its original value.

We ran a smaller experiment than the standard Waterbirds parameters we described in Appendix B. Namely, our seed set was of size 32, we acquired 64 examples on top of that, and we acquired one example at a time. We also defined the validation accuracy as having plateaued the fifth time it did not increase.

We ran twelve paired experiments where for twelve different randomized seed sets, we performed the Waterbirds experiment four times - with random sampling and stopping when validation accuracy plateaued, with random sampling and stopping when training loss decayed, with uncertainty

sampling and stopping when validation accuracy plateaued, and with uncertainty sampling and stopping when training loss decayed.

We found that these experiments achieved:

1. Random sampling + Stop from validation accuracy: average accuracy = 69.11%, average duration =  3.5 hours

2. Random sampling + Stop from training loss: average accuracy = 69.93%, average duration = 38 minutes

3. Uncertainty sampling + Stop from validation accuracy: average accuracy = 83.64%, average duration =  4.5 hours

4. Uncertainty sampling + Stop from training loss: average accuracy = 85.62%, average duration = 74 minutes

Thus, we concluded that the by stopping our finetuning step just by waiting for the training loss to decay to 0.001 of its original values, we could achieve comparable if not better accuracies, spend a fraction of the time, and remove the need for a labeled validation set.

## E    VISION TRANSFORMER MODEL

Results for Treeperson when using ViT-B/16 are presented in Figure 8. While active learning improves upon random sampling, the gains are not as large as for BiT, perhaps because ViT was pretrained for almost 8x fewer epochs.

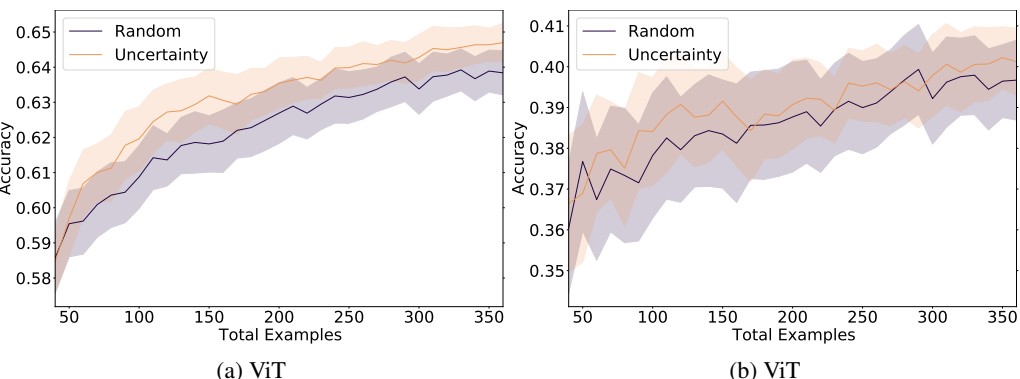

(a) ViT                                    (b) ViT

Figure 8: **Active learning improves upon random sampling for Treeperson when using ViT-B/16.** However, the gains are not as large as for BiT, perhaps because ViT was pretrained for almost 8x fewer epochs.

## F    DOSE RESPONSE

Results for the Waterbirds dose-response experiment are presented in Figure 10. Even when the training and test distributions are the same, active learning allows the pretrained BiT model to outperform random sampling. However, as the train and test distribution diverge, the benefit of active learning increases, indicating the importance.

## G    EFFECT OF MODEL SCALING AND PRETRAINING ON ACQUISITION

For the Waterbirds model scaling experiment, we track the examples each model acquires. These are presented in Figure 11. The acquisition patterns of all the pretrained models look fairly similar—they upsample both minority (landbird/water-background, waterbird/land-background) subclasses. However, the non-pretrained model is unable to capture that distinction, and is only able to upsample images with a water background, resulting in worse performance. A summary of final class acquisition ratios is available in Figure 12.

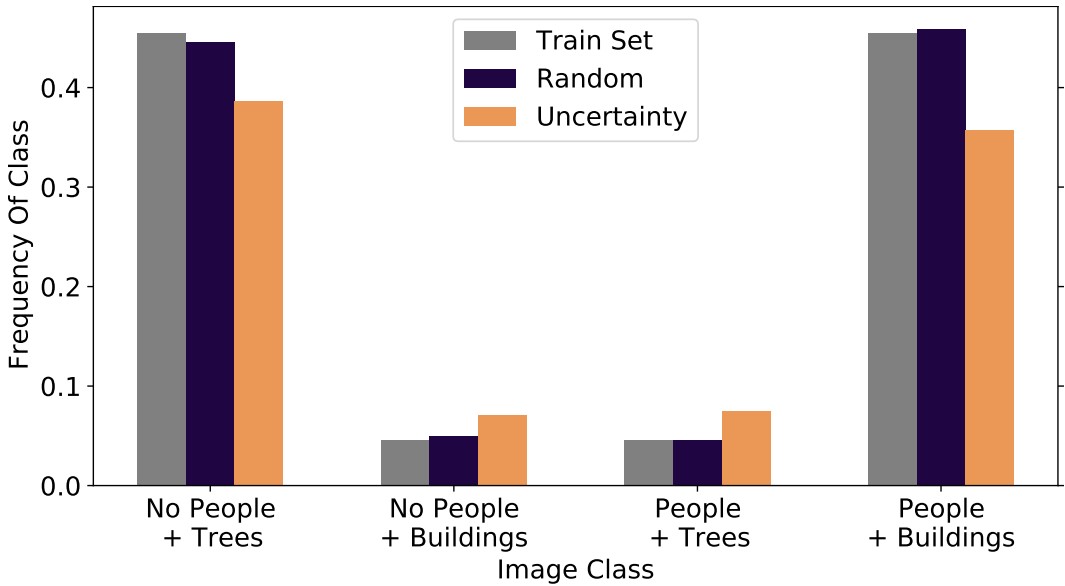

Figure 9: **Acquisitions for Treeperson using ViT.** The ViT model requested labels for minority classes at a significantly lower rate than did the BiT model.

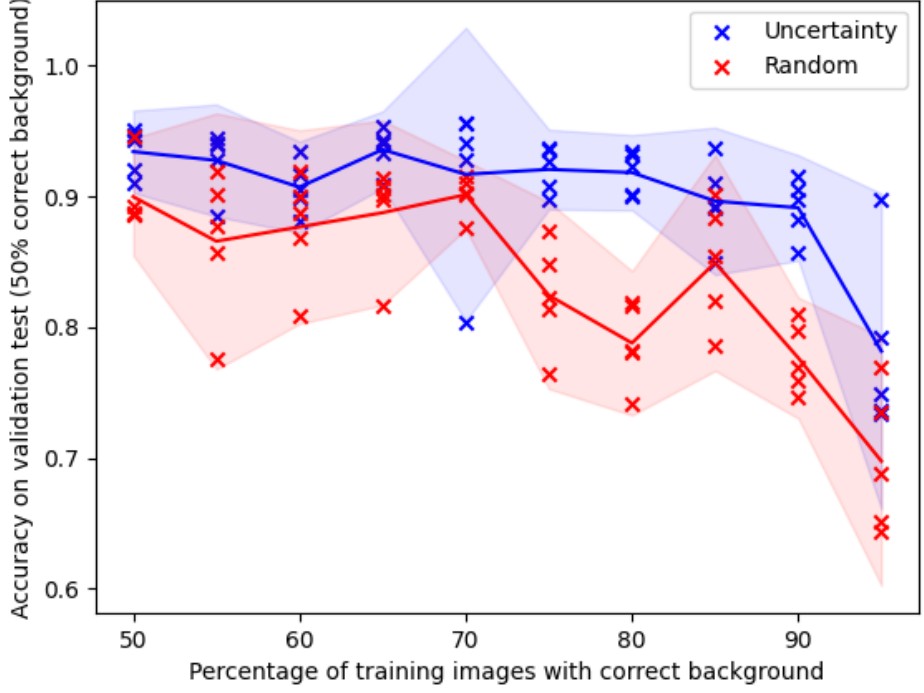

Figure 10: **Waterbirds background mismatch dose-response experiment.** As the train and test distribution diverge, the benefit that active learning provides increases.

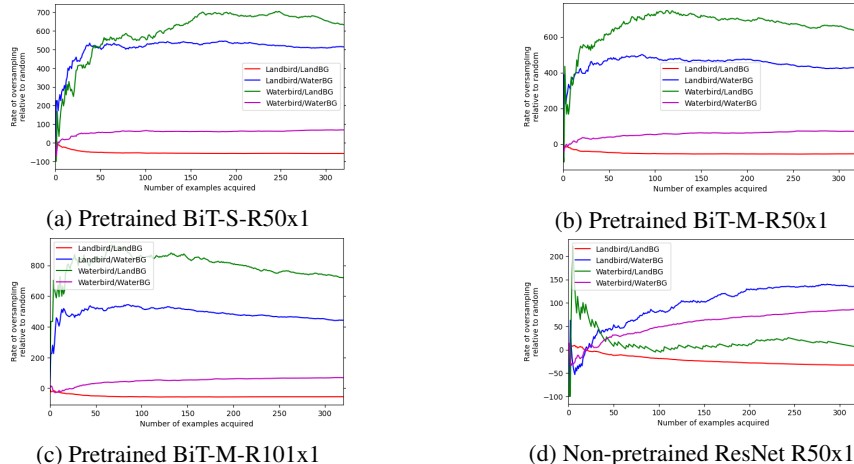

(a) Pretrained BiT-S-R50x1  (b) Pretrained BiT-M-R50x1

(c) Pretrained BiT-M-R101x1  (d) Non-pretrained ResNet R50x1

Figure 11: **All pretrained models acquire disambiguating subgroups much more efficiently than their non-pretrained counterparts.** The pretrained models do not simply oversample based on the the bird or the background; instead they oversample disambiguating examples which have mismatched backgrounds. By contrast, the non-pretrained model only oversamples images with a water background, and accordingly is less able to perform well on the balanced validation set.

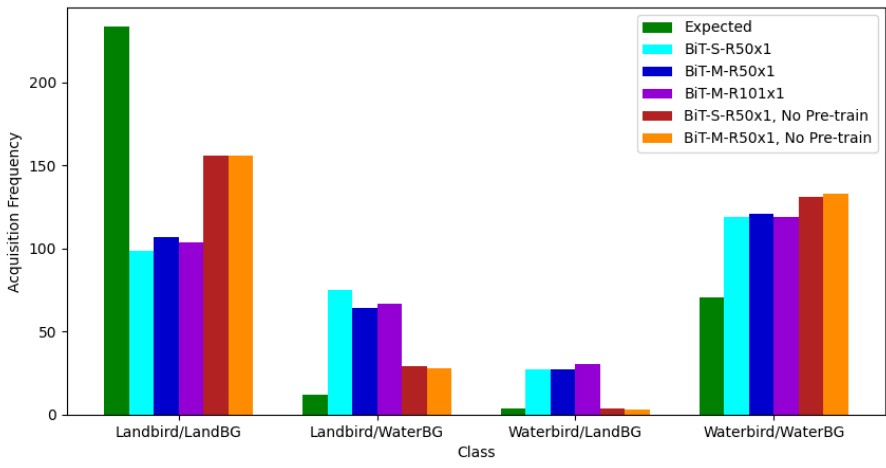

Figure 12: **Pretrained models ask for labels of images with mismatched backgrounds, while non-pretrained models do not.**

## H   Visualization Of Image Embeddings Of Pretrained Bit-M

As seen in Figure 13b, when performing t-SNE on the Waterbirds image embeddings, we find that the t-SNE of the pretrained model has much more structure than that of the unpretrained model. In the unpretrained model, the image embeddings of the landbird/land background, landbird/water background, and waterbird/water background classes are distributed uniformly about the center of the t-SNE projection. However, we observe much more structure in the pretrained model's t-SNE. In particular, the most noticible difference is that the landbird/water background and waterbird/land background classes can be found near the other landbird and waterbird images, respectively. This shows that even without any finetuning, the pretrained model already has learned features to help process the type of bird that appears in the image.

In the t-SNE plots, the black stars represent the embeddings of the first 10 images that were acquired by each model. For the pretrained model, the acquired examples very clearly fall along the waterbird-landbird boundary in the projected feature space. However, in the unpretrained model, the acquired examples are distributed randomly. This suggests that the features acquired during pretraining enables them to select examples that fall near the decision boundary of a new unseen task.

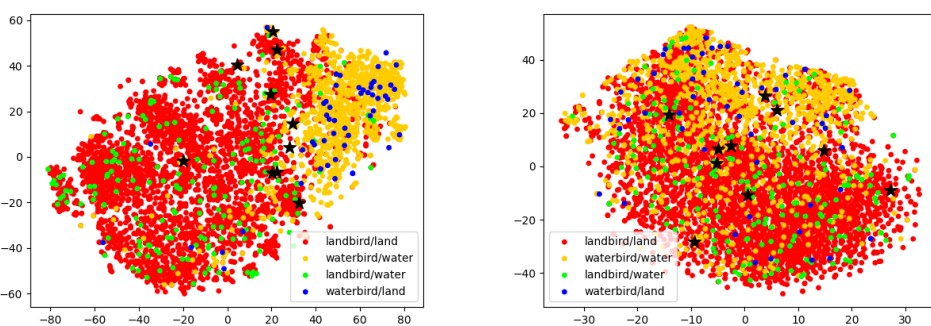

(a) t-SNE of Waterbirds image embeddings from pretrained Bit-M

(b) t-SNE of Waterbirds image embeddings from non-pretrained Bit-M

Figure 13: **Without any finetuning, pretrained models already embed images into a useful feature space.**

## I   ROLE OF LABEL IMBALANCE

We attempt to isolate the role of label imbalance by investigating CIFAR-10 (Krizhevsky, 2009), a widely-used balanced dataset where task ambiguity is not known to be a common problem. We train uncertainty- and random-sampling models on both the original CIFAR-10 dataset, as well as an imbalanceed variant where half of the classes have 90% of their examples removed. As shown in Figure 14, random sampling slightly outperforms uncertainty sampling on the original CIFAR-10 dataset, but uncertainty sampling performs better in unbalanced settings, both when the validation dataset is imbalanced as well as when it is balanced.

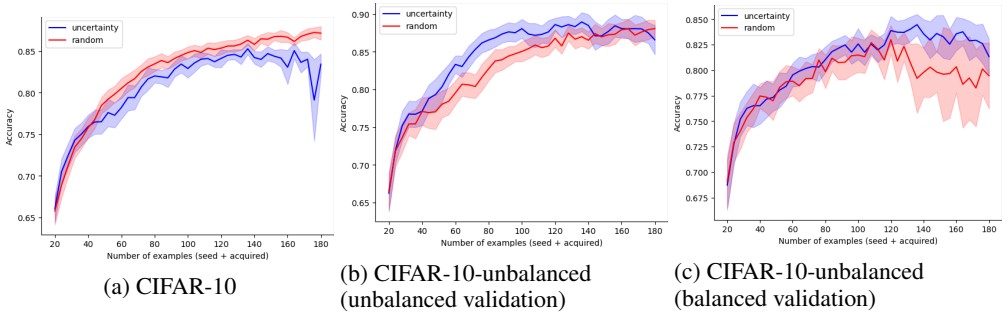

(a) CIFAR-10

(b) CIFAR-10-unbalanced (unbalanced validation)

(c) CIFAR-10-unbalanced (balanced validation)

Figure 14: Accuracy on CIFAR10 as more samples are acquired with uncertainty vs random acquisition. Shaded regions represent 95% CIs (Gaussian approx.). All runs are with the pretrained model BiT-M-R50x1. (a): The usual CIFAR10. (b) and (c): CIFAR10 where the training set has 5 out of 10 classes from which 90% samples are removed. (b) is accuracy on validation split with the same distribution as train. (c) is on balanced validation split.

