# OpenReview forum: "Pretrained models are active learners"
_ICLR.cc/2022/Conference — ICLR 2022 Submitted_

### Official Review · Reviewer_Gazs · 2021-10-24

**Correctness:** 4
**Technical Novelty And Significance:** 2
**Empirical Novelty And Significance:** 2
**Recommendation:** 3
**Confidence:** 4

**Main Review:**

Strengths:

This paper is clearly written. It describes experiment setup and results very clearly. The advantage of pretraining is also clearly established.

Weaknesses:

For a purely empirical paper (which is totally fine for me), that the depth of investigation is a bit lacking. I would like to see more comprehensive results to be convinced that the performance gap is indeed due to pre-training vs. no pre-training, instead of certain hyper-parameter choices happen to play well with the pre-trained models. Some specific items that I would encourage the authors to provde are:

1. More acquisition functions should be studied. While uncertainty-based acquisition function is perhaps the most popular, many others have been recently proposed. For a recent list, see the related work section of [1].

2. It is not clear which part of the pre-training is most helpful for active learning. It is known that pre-trained models are very efficient at learning from few examples, but they seem to struggle here, especially in Fig. 2, where the random baseline makes very small progress even with hundreds of data points (which seems to be rare for pre-trained models). I suspect that this is at least partly due to the issue below, i.e. the mismatch between training and test distribution, so some clarification would be appreciated.

In addition, there seems to be a disconnect between the setup described in Fig. 1 and the actual experiments conducted. In the Fig. 1 setup, the existing (small) training set is not sufficient to disambiguate among a set of candidate functions, so the promise of active learning is to find the maximally confusing data point from the unlabeled set and thus achieve efficient disambiguation. Notably, the training and test distributions do not *need* to be different. However, in the experiments, in all three settings, the training and test distributions are intentionally different, so that the models that have undesirable working mechanisms, e.g. relying on spurious correlation, are "defined" to be bad. Is this a characterization of the datasets used? If so, it seems that here, active learning with pre-trained models are better not in the typical sense of improving the performance on the training task, but instead helping the model to gain some other behaviors.

Interestingly, [1] shows that the optimal acquisition strategy (w.r.t. the training task) seems to sample quite homogeneously in the input and output space, and a uniformity regularization would actually improve existing acquisition functions. By comparison, Fig. 6 show strong mismatch between the acquired sample distribution and the data distribution. Given that the paper evaluates the "other behavior" performance, if my understanding in the paragraph above is correct, I do not find it surprising. Nevertheless, I would like to encourage the authors to experiment with the uniformity regularization (Sec. 6.5 of [1]) and discuss any findings and relationships.

[1] Towards Understanding the Behaviors of Optimal Deep Active Learning Algorithms, AISTATS 2021

Minor issues:

1. Images should be produced as PDF files and embedded into the paper. This ensures that the image is saved in the vector format, rather than bitmap. For matplotlib, this is as easy as changing the suffix from png/jpg to pdf, i.e. `plt.savefig('figure.pdf')`.

2. The texts on most figures are too small. In general, they should approximate match the text size, or at least the footnote size, but some of them are barely readable when printed out.

3. A reference to [2] is recommended in footnote 1 when discussing calibration, since it is the original paper that investigates the confidence calibration problem of neural networks.

[2] On Calibration of Modern Neural Networks, ICML 2017.

**Summary Of The Paper:**

This paper investigates the active learning performance of pre-trained models vs their non-pre-trained counterparts on both vision and NLP tasks. Specifically, the investigation focuses on datasets with spurious correlation, domain shift, and label imbalance. Empirical results generally show that the pre-trained models with the uncertainty acquisition function performs much better than the random baseline and their un-pre-trained counterparts.

**Summary Of The Review:**

Overall, I think this paper is on track to make a good contribution, but would need a more rounded-out experimental execution. Therefore, I am giving a reject recommendation for now, but I'd be happy to increase my score if the authors provide the requested revision.

---

> ### Author Response · Authors · 2021-11-23
> **Response**
>
> We appreciate R4's time and effort reviewing the paper, that they found the work to be "clearly written" and that we "clearly established" the effect of pretraining. We also appreciate the reviewer expressing their willingness to accept the paper if their requested revisions were made. We address each of them here:
>
> **More acquisition functions should be studied.**
>
> We have added results for entropy and margin sampling (see general comment). These results bolster the generality of our work—we appreciate the suggestion!
>
> **Why does the random baseline make small progress even with hundreds of datapoints? + Relation to train-test mismatch.**
>
> This is correct! Pretraining is no cure-all, and we believe applications that feature task ambiguity (which may manifest through distribution shift) are one of the most important outstanding challenges. To capture the precise effect of these shifts, we conduct a "dose-response" experiment where we vary the fraction of mismatched examples in Waterbirds from 5% to 50%. See the main comment for more details.
>
> **"In the experiments, in all three settings, the training and test distributions are intentionally different, so that the models that have undesirable working mechanisms, e.g. relying on spurious correlation, are "defined" to be bad. Is this a characterization of the datasets used?"**
>
> Yes! Our core focus is active learning as a means to resolve task ambiguity. Real world datasets exhibit a range of phenomena that make specifying the desired task challenging (e.g. spurious correlations, latent subgroups, etc). We conduct experiments on four diverse datasets where task ambiguity is salient to test whether active learning helps the model acquire the correct behavior.
>
> **Minor issues**
>
> We have incorporated these into this newest revision!
>
> **Overall, I think this paper is on track to make a good contribution, but would need a more rounded-out experimental execution**
>
> In addition to the experiments mentioned in the review, we have added additional experiments to flesh out other aspects of our paper in additional detail (see general comment).
>
> We thank the reviewer again for their helpful comments! We kindly ask them to consider recommending an acceptance if they feel most of their concerns have been resolved and the ICLR community would benefit from reading the work.

---

### Official Review · Reviewer_uqL6 · 2021-11-01

**Correctness:** 3
**Technical Novelty And Significance:** 2
**Empirical Novelty And Significance:** 3
**Recommendation:** 5
**Confidence:** 5

**Main Review:**

I find the paper and the direction to be interesting and I think the paper is well written and well motivated. However, a major problem the paper faces is that there is not enough experiments in the paper to validate the claims made by the authors.

In the experiments, the authors show the results of pre-training on all the datasets they used and it is clear that active learning performs better than random sampling however on experiments comparing pre-trained models to un-pretrained ones, the authors only show experiments on two datasets. They report a failure case for one dataset in which the model performance is not distinguishable for neither pre-trained or un-pretrained models.

Focusing on the two experiments, in Figure 8, active learning actually does perform better than random sampling even if the relative performance on the experiment is not as good as that of the pre-trained models. In addition, from the plots active learning performance is indistinguishable to random performance on Pretrained BiT-S-R50x1 and only gets marginally better on the other pre-trained models as more examples are added.
On both Figure 7 and 8, we observe that random sampling on pre-trained models outperforms active learning on the un-pretrained counterpart. A possible explanation is that the model has only seen few examples to learn good features that are representatives of the class. This explains why pre-training does better since it is trained on a large pool of unlabeled data and fine-tuned on the specific task hence it learns richer/better feature for model training.

Given the above limitations, it is hard to verify the main claims of the paper. Additional experiments on different datasets including text data will be needed. Also, it will be helpful to conduct experiments in settings where the un-pretrained models performs reasonably well on the task when trained with just the seed labels. Lastly, it is important to consider different architecture models for the pre-trained models and not just variants of the BiT model.

Minor suggestion:
In low-shot settings, entropy sampling may be a better query strategy for the un-pretrained model.

**Summary Of The Paper:**

The authors set out to investigate if active learning is an emergent property of pre-training. That is if running active learning with pre-trained models gives better result than using the same models without pre-training. They run several experiments on different text and image datasets first showing that active learning performs better than random sampling on pre-trained models and secondly that pre-trained models perform better than un-pretrained ones for active learning.

**Summary Of The Review:**

It's an interesting paper but the authors need more empirical validation to establish some of the claims. See detailed comments in main review.

---

> ### Author Response · Authors · 2021-11-23
> **Response**
>
> We appreciate R3's time and effort reviewing the paper, and that they found the work to be "interesting…well written and well motivated." We respond to the key points raised:
>
> **Additional experiments to flesh out the generality of the method (additional architectures, sampling methods, and results for unpretrained models)**
>
> Thank you for these suggestions! We have conducted additional experiments to address each of these three points, and believe they significantly enhance the strength of the paper. See the general comment for more details.
>
> **"They report a failure case for one dataset in which the model performance is not distinguishable for neither pre-trained or un-pretrained models…active learning performance is indistinguishable to random performance on Pretrained BiT-S-R50x1 and only gets marginally better on the other pre-trained models as more examples are added**
>
> If this refers to the iWildCam dataset results, we believe this is an incorrect interpretation of the data. These results show that active learning is an emergent property of pretraining: active learning hurts for unpretrained models, has neutral effect on the BiT-S-R50x1 model (pretrained on less data), and has a very large (+8%) improvement on the BiT-M-R50x1 and Bit-M-R101x1 models (pretrained on more data). While gains do level off towards 90% accuracy, even strong pretrained models typically level off at some ceiling performance, so this shouldn't be interpreted as a failing of the method.
>
> **"on both Figure 7 and 8, we observe that random sampling on pre-trained models outperforms active learning on the un-pretrained counterpart…pre-training does better since it is trained on a large pool of unlabeled data and fine-tuned on the specific task hence it learns richer/better feature for model training."**
>
> Correct! We believe the richer representations learned by pretraining not only enable better general performance but also enable better active learning: choosing informative examples which improve performance above random sampling. We substantiate this claim through both increased finetuning accuracy for AL with pretrained models, as well as by demonstrating that *only* AL with pretrained models chooses the informative examples. We also have some new experiments demonstrating this through probing model representations with linear classifiers and t-SNE visualizations. See the general comment for more details and discussion.
>
> We thank the reviewer again for their helpful comments, and kindly ask them to consider recommending an acceptance if they feel most of their concerns have been resolved and the ICLR community would benefit from reading the work.

---

### Official Review · Reviewer_bL9o · 2021-11-06

**Correctness:** 3
**Technical Novelty And Significance:** 2
**Empirical Novelty And Significance:** 2
**Recommendation:** 3
**Confidence:** 4

**Main Review:**

To begin with, these are interesting observations and likely have practical implications. The strength of the paper are the empirical reductions in sample complexity and increase in performance. However, in my assessment, these results are fairly preliminary. While in two domains, the image classification results are clearly stronger than the text classification results (but no real explanation for the reason — I guess it is that the ‘task ambiguity’ is consistent from train to test (?)). If this is the hypothesis, it seems that one could perform a ‘dose-response’ like analysis of background shift prevalence between train/test and characterize the strength of this relationship, which seems important as this is the primary justification of why AL with pretrained models is more powerful. I have a potentially simpler hypothesis, that these improved representations remove a lot of the cold-start problems with AL that have previously been mitigated with pre-clustering, careful seed selection, etc. Basically, the AL selection can focus on fitting the conditional distribution associated with classification and not be concerned with the joint distribution of cold-start problems. The compressed representation has lower variance within coordinates and selecting ‘difficult’ examples is more likely to be fruitful (i.e., pathological cases are mitigated). Of course, I am just speculating and my hypothesis is neither orthogonal nor contradictory to the one proposed — but there is no careful empirical analysis or any analytical analysis to justify the proposed hypothesis either.

On a more practical level, I would also expect more AL querying functions as this is largely an empirical papers and I see no reason why uncertainty sampling is guaranteed to be superior (although it is intuitively appealing that pathological cases probably disappear).

From a scholarly perspective, there has been recent related work at least in the NLP space (which I am more familiar with):
[Yuan, Lin & Boyd-Graber, Cold-start Active Learning through Self-supervised Language Modeling, EMNLP20]
[Margatina, Barrault & Aletras, Bayesian Active Learning with Pretrained Language Models, 2021]
[Shelmanov, et al., Active Learning for Sequence Tagging with Deep Pre-trained Models and Bayesian Uncertainty Estimates, EACL21]
There are more, but this gives some good seeds to follow-up on.

Overall, the writing is clear, but I would recommend a few things: (1) put an algorithm somehwere to verify if you are fine-tuning the embeddings or the overall model and bring a bit more of the model details into the paper from the appendices and (2) change the title as it is misleading; something like “Pretrained Models Make Good Active Learners”.

In summary, the preliminary observations are interesting and notable in some cases. There are enough ‘additional’ experiments to demonstrate that the authors have a promising path to a theory. However, to be a more impactful finding, I lean toward better contextualization, more experiments that vary the active learning querying function, and more empirical/analytical results to support a clear theory. Interesting, but I don’t believe ready for acceptance in ICLR.


**Summary Of The Paper:**

The authors describe interesting empirical observations regarding using uncertainty sampling to select examples to fine-tune models that use pretrained embeddings and provide some hypotheses regarding the reasons for these performance improvements. Specifically, (1) from a methodological perspective, they propose using uncertainty sampling (i.e., least confident selection) to select examples for fine-tuning image/NLP pretrained models and (2) from an empirical perspective, they use Waterbirds/Treeperson/iWildCam2020-WILDS for image classification and Amazon-WILDS for review star prediction based on text and compare with random sampling — noting that these are settings where there is known covariate shift between train/test with semantic meaning to induce interpretable spurious associations (e.g., background in images). The proposed method works overall, especially on the image datasets, and they also dig into the types of examples selected — noting that they align with expected ‘difficult’ examples (depending on the setting).

**Summary Of The Review:**

The authors make some interesting and potentially impressive observations regarding the performance of ‘simple’ active learning (i.e., uncertainty sampling) in the context of pre-trained models — showing positive results both on image and text classification problems. Additionally, they show that biasing toward difficult examples for selection may be correlated with being able to ignore spurious features (e.g., background in images). However, the experiments are limited to ‘least confidence’ uncertainty sampling, there is non-negligible missing contextualization wrt related work, and there isn’t a strong justification for the observed performance improvements. Thus, my assessment is that this submission should be rejected in its current form.

---

> ### Author Response · Authors · 2021-11-23
> **Response**
>
> We appreciate R2's time and effort reviewing the paper, and that they believe "these are interesting observations and likely have practical implications." We respond to the key points raised:
>
> **More AL querying functions**
>
> We have added results for entropy and margin sampling (see general comment). These results bolster the generality of our work—we appreciate the suggestion!
>
> **Analytic support for hypothesis (e.g. dose response experiment)**
>
> We conducted such a dose-response analysis with the Waterbirds dataset (see general comment) demonstrating that the fraction of mismatched examples (e.g. landbird on water) is a key factor influencing the success of our method, as expected. Thank you for the suggestion! See the general comment for more details.
>
> **What explains the strength of image results over text results? Is this due to reduced task ambiguity?**
>
> Yes—comparing Figures 5 and 6, we see the sampling ratios between the majority and minority classes are more extreme for the image datasets than the text dataset, suggesting either that the shift is less extreme or that the RoBERTa model is less adept at identifying informative examples than the image models we considered.
>
> **Contextualization with respect to prior work**
>
> We thank the reviewer for those links to related work. We have updated our related work section, which now totals a page in length. We certainly agree that pretrained models have been used in many recent AL works, often motivated by informal hypotheses that pretrained models might help (e.g. Yuan et al argue that cold-start AL is challenging because typical deep models are poorly calibrated, and uses the pretraining loss as a proxy for uncertainty). We build on these works in two key ways:
> 1) We prove out these intuitions empirically by demonstrating that pretrained models actually do make better active learners by comparing to unpretrained baselines
> 2) We show that active learning can choose examples that resolve task ambiguity or distribution shift, resulting in large improvements on a range of real-world tasks across modalities, model families, and acquisition functions
>
> **"verify if you are fine-tuning the embeddings or the overall model"**
>
> Confirming that we use the standard finetuning approaches used for RoBERTa, BiT, and ViT—namely finetuning all the parameters (including embeddings, in the case of transformer models).
>
> We thank the reviewer again for their helpful comments, and kindly ask them to consider recommending an acceptance if they feel most of their concerns have been resolved and the ICLR community would benefit from reading the work.

---

> > ### Comment · Reviewer_bL9o · 2021-12-02
> > **response to rebuttal**
> >
> > Thanks much for the rebuttal (and submission). First of all, I did read the updated version of this submission and do think it is a better paper and do appreciate the 'dose-response' experiment. In my estimation, this submission is now somewhere between a marginal and borderline reject, but I still think it is a reject. Specifically, there are two issues I continue to observe: (1) I am still not convinced regarding how much of the gains are due to active learning vs pre-training with respect to different cases and (2) there are many observations here without a clear narrative or theory (which may have actually gotten worse with the additional experiments). Basically, I think the authors have the pieces of a good work, but need to more clearly state their hypotheses and support them more unequivocally (or develop experiments to explain the cases where their approach doesn't work as well). It's a solid technical report and has some good findings, but isn't quite a strong research paper yet (in my opinion).

---

### Official Review · Reviewer_KGpV · 2021-11-08

**Correctness:** 4
**Technical Novelty And Significance:** 3
**Empirical Novelty And Significance:** 3
**Recommendation:** 8
**Confidence:** 3

**Main Review:**

I wanted to see two additional directions of investigations.
1. First, does the choice of active learning heuristic matter? My intuition is that it probably does not, but it would be nice to see a comparison between uncertainity sampling heuristic and another information seeking heuristic such as entropy sampling. The direction of investigating other active learning techniques (such as query-by-committe or expected model change) also remain open.

2. All the pre-trained models considered are supervised pre-trained models. What happens if we use a large, unsupervised pre-trained model (for example a masked language language model)? Do the same results hold true?


**Summary Of The Paper:**

This paper investigates if using large, pretrained models in an active learning setup helps achieve better performance with lesser data when compared to using randomly sampled data. In order to conduct this investigation the authors study the empirical performance of large pre-trained models on some image datasets and a text dataset. In both cases large pre-trained model is finetuned on a small amount of seed data and then an active learning procedure is used (in this paper the AL procedure is an uncertainty sampling procedure) to collect more data. The datasets are chosen to illustrate several conceptual issues (i) distinguishing causal from spurious correlations (ii) measuring robustness to distribution shifts (iii) role of data imbalance.

Experiments are performed to show that using an active learning procedure indeed helps improve performance using only a small amount of actively labeled training dataset.  The paper is well written and the results are convincing and insightful.

**Summary Of The Review:**

I like the key question and the experimental setup of this paper and would recommend an accept. It would have been better had the authors considered another AL method, along with uncertainty sampling to investigate the relative impact of pre-training and the active learning methodology in data savings.

---

> ### Author Response · Authors · 2021-11-23
> **Response**
>
> We appreciate R1's time and effort reviewing the paper, that they liked "the key question and experimental setup of this paper," and that they recommended accepting the work! We address the additional directions the reviewer suggested:
>
> **Choice of active learning heuristic**
>
> Thanks for this suggestion. We have added additional experiments for entropy sampling and margin sampling. See the general comment for more information about these experiments as well as several others. We believe these experiments broaden our method's demonstrated generality, and appreciate the suggestion.
>
> **"What happens if we use a large, unsupervised pre-trained model (for example a masked language language model)"**
>
> This was a question that interested us as well! Kindly note that RoBERTa, used in the Amazon experiments, is a masked language model.
>
> We hope these additional results, along with those in the general comment, bolster the reviewer's confidence in their recommendation.

---

### Author Response · Authors · 2021-11-23
**General Comment**

Sincere thanks to the reviewers for their time and thoughtful reviews! We appreciate that reviewers found our work to be an "interesting direction" that was "well written and well motivated" and was "likely to have practical implications."

We also appreciate that reviewers broadly understood the specific details of our experimental procedures and results—namely that significant gains from uncertainty sampling exist when using pretrained models, but that these gains are absent or even negative without the use of pretraining.

We wish to uplift a few higher-level points and present additional experiments (which also appear in the draft):
## We address an important and underexplored problem
Our work considers a fundamental problem in machine learning: translating a desired task into an unambiguous *specification* of that task via a set of training examples.

This is challenging because available datasets frequently contain hard-to-detect skews, spurious correlations, shortcuts, imbalances, or other kinds of ambiguity that only become apparent during deployment when the data distribution shifts (see [0] for a survey).

Here, we investigate active learning as a simple way to elicit training examples from users that will induce something closer to the true task in a model. Thus, we treat active learning not simply as a black box for improving model performance, but as an interpretable procedure that should choose specific examples that resolve ambiguity in cases where we can identify it.

We believe the problem of task ambiguity is underexplored despite its conceptual and practical importance, especially given the recent rise in pretrained models and research on few-shot learning where task ambiguity is especially salient.
## Our solution is widely applicable and the effect we find is large!
Our core result is that active learning empirically does help here, by what was frankly a surprisingly large amount to us (gains of +12, +2, +2.5, and +8 points) with even larger gains on some minority classes. We find these gains striking, and perhaps underappreciated in the reviews, especially given the widely-documented challenges of applying active learning in practice [1, 2]. Moreover, these gains resulted from a single active learning recipe that was not tuned across tasks or models and does not require the use of a validation set, suggesting its broader applicability to practitioners.
## Pinpointing the impact of task ambiguity with a "dose-response" experiment
Several reviewers wonder whether active learning is actually helping the model identify the true task (resolving task ambiguity) or whether pretraining simply helps ameliorate the cold start problem for active learning via a better feature space. To investigate, we construct alternate versions of the Waterbirds dataset with varying levels of task ambiguity. The official Waterbirds dataset has 95% of examples with matched backgrounds, and 5% mismatched. We use the provided Waterbirds code to construct versions of the dataset with 50% to 95% shift, holding all other aspects constant. We find significantly larger gains on versions of the dataset with smaller fraction of mismatched examples (gains average ~4% between 50-70% shift, and ~10% between 75-95% shift). This provides stronger evidence that active learning is actually helping the model identify the true task, as opposed to pretrained feature spaces simply making active learning work better.
## Analysis of the feature space
To gain further insight into why pretraining helps, we train linear probes for each (bird, background) pair on the pretrained model's representations for Waterbirds, after training on a seed set, and after acquiring 20 examples with AL. We find that the pretrained model has representations which better distinguish each of the four subtypes vs the unpretrained model, and that these representations are further improved after training on the seed set and acquired examples. This provides additional evidence that pretraining helps because it enables the model to better identify and select examples based on preexisting high-level features.

We also visualize BiT's second-to-last layer embeddings for the Waterbirds dataset, *without any finetuning* using t-SNE. We find that the pretrained model exhibits far more separation by class, suggesting it is better able to identify useful features in the input even before training. These conclusions help further characterize how pretraining provides a feature space more amenable to active learning.

[continued...]

---

> ### Author Response · Authors · 2021-11-23
> **General Comment [cont.]**
>
> ## Additional experiments to validate the generality of our conclusions
> The main request from reviewers was additional experimental results to validate that our conclusions hold across a wider range of active learning methods and architectures:
> - *Two additional active learning methods*: entropy and margin sampling. These are equivalent to least confidence sampling for binary classification tasks, thus we experiment on the 182-class iWildCam dataset. All methods consistently outperform random sampling, with margin sampling often outperforming both least confidence and entropy. This demonstrates our results are not specific to least confidence and that other sampling strategies may enable further gains.
> - *Other models in addition to BiT and RoBERTa*: We consider a moderate-size vision transformer model (https://huggingface.co/google/vit-base-patch16-224), expanding our coverage of computer vision architectures to the other prominent model family currently in use. On Treepereson, we see small but consistent gains and oversampling of minority classes, similar to BiT. The oversampling ratio and gains relative to random sampling are of smaller magnitude, perhaps reflecting the fact that the vision transformer was pretrained on ImageNet-21k for far fewer epochs then BiT (9 vs 70).
> - *Results for unpretrained models on Treeperson*: Unpretrained BiT-M with uncertainty sampling yields comparable or worse runs to random sampling. This provides an additional point of evidence that pretraining is crucial here.
> - *Do hyperparameters only "play nice" with pretrained models?*: We perform experiments on Waterbirds with six different settings of learning rates and early stopping loss ratios. In all cases, we find random sampling matches or outperforms uncertainty sampling for unpretrained models. This provides additional evidence that active learning doesn't simply fail in unpretrained models due to unfavorable hyperparameters.
>
> ## Updated draft
> We have added these results to the draft, and also redesigned the figures to be higher resolution and easier to understand and read. Following the suggestion of a reviewer, we have also changed the title of the paper to "Pretrained Models are Good Active Learners."
>
> [0] Robert Geirhos, Jörn-Henrik Jacobsen, Claudio Michaelis, Richard Zemel, Wieland Brendel, Matthias Bethge, Felix A. Wichmann. Shortcut Learning in Deep Neural Networks
> [1] David Lowell, Zachary Chase Lipton, and Byron C. Wallace. Practical obstacles to deploying active learning.
> [2] Siddharth Karamcheti, Ranjay Krishna, Li Fei-Fei, and Christopher D. Manning. Mind your outliers! investigating the negative impact of outliers on active learning for visual question answering
> [3] Goh, Gabriel, Nick Cammarata, Chelsea Voss, Shan Carter, Michael Petrov, Ludwig Schubert, Alec Radford, and Chris Olah. Multimodal neurons in artificial neural networks.
> [4] Shrey Desai and Greg Durrett. Calibration of pre-trained transformers
> [5] Dan Hendrycks, Kimin Lee, and Mantas Mazeika. Using pre-training can improve model robustness and uncertainty

---

### Author Response · Authors · 2021-11-29
**Follow up before end of discussion period today**

Dear reviewers,

Thank you all again for the time you spent writing your reviews! We know our rebuttal might be on the longer side, but we wanted to give thorough answers to your questions and detail the results of the experiments you requested.

If you feel you have any remaining concerns which might preclude an acceptance, please let us know as soon as possible so we may respond if appropriate. If you feel most of your concerns have been addressed, we'd kindly ask you to update your scores in light of the changes. Thank you again, and happy holidays to any who are celebrating!

---

### Decision · Program_Chairs · 2022-01-20

**Decision:**

Reject

**Comment:**

The paper shows that active learning is an emergent property of pre-trained models. They show that simple uncertainty sampling improves sample efficiency by 6 times (up to 6x fewer samples for the same accuracy). This is an interesting and important observation that has practical implications.

Initially, there were various concerns regarding the message of the paper, including the tile and use of uncertainty function in AL and lack of enough experiments that were addressed through rebuttal period.

However, there are still remaining concerns that lead to the paper not being ready for publication. Namely,
  (1) Clear discussion on how much of the gains are due to active learning vs pre-training with respect to different cases. it is also worth investigating additional causes for the failure cases.
  (2) there are many observations here without a clear narrative or theory.
Moreover, making the story more cohesive will strengthen the paper.